# Bacterial Cellulose and Emulsified AESO Biocomposites as an Ecological Alternative to Leather

**DOI:** 10.3390/nano9121710

**Published:** 2019-11-29

**Authors:** Marta Fernandes, António Pedro Souto, Miguel Gama, Fernando Dourado

**Affiliations:** 12C2T-Centre for Textile Science and Technology, University of Minho, Campus de Azurém, 4800-058 Guimarães, Portugal; marta.fernandes@det.uminho.pt (M.F.); souto@det.uminho.pt (A.P.S.); 2CEB-Centre of Biological Engineering, University of Minho, Campus de Gualtar, 4710-057 Braga, Portugal; fdourado@deb.uminho.pt

**Keywords:** bacterial cellulose, acrylated epoxidized soybean oil, biocomposite, emulsion, exhaustion, alternative leather

## Abstract

This research investigated the development of bio-based composites comprising bacterial cellulose (BC), as obtained by static culture, and acrylated epoxidized soybean oil (AESO) as an alternative to leather. AESO was first emulsified; polyethylene glycol (PEG), polydimethylsiloxane (PDMS) and perfluorocarbon-based polymers were also added to the AESO emulsion, with the mixtures being diffused into the BC 3D nanofibrillar matrix by an exhaustion process. Scanning electron microscopy (SEM) and Fourier transform infrared (FTIR) spectroscopy analysis demonstrated that the tested polymers penetrated well and uniformly into the bulk of the BC matrix. The obtained composites were hydrophobic and thermally stable up to 200 °C. Regarding their mechanical properties, the addition of different polymers lead to a decrease in the tensile strength and an increase in the elongation at break, overall presenting satisfactory performance as a potential alternative to leather.

## 1. Introduction

The tannery industry faces several challenges associated with high environmental impact, scarcity of raw materials and increasing consumer demand for environmentally friendly products. The worldwide production of leather is approximately 20 billion square feet per year [1]. To produce 1 ton of leather, 6.7 tons of raw skin [2], 57,000 liters of water [3], and 3.35 tons of chemicals are required [4]. Worldwide, for bovine skin, 370 billion liters of water are consumed annually, generating 6.5 million tons of solid waste. This research intends to contribute to the reduction of the animal hide dependency by the development of composites from bacterial cellulose (BC) as structuring material and activated vegetable oils as a flexibilizing, mechanical reinforcing and hydrophobizing agent. BC is a biopolymer produced by bacterial fermentation that consists exclusively of a three-dimensional structure of pure cellulose nanofibers. Chemically, BC is identical to vegetable cellulose but the nano-scale of its fibers offers a significantly higher surface area [5].

Regarding its application in the textile and footwear sectors, the first proof of concept of the use of BC as an alternative to leather emerged in the 1990s, in the Philippines [6]. In the last decade, the designer Suzanne Lee has expanded the possibility of using BC in the manufacture of clothing and footwear, by resorting to the handmade production of BC, which is washed, adjusted to a predefined form, dried and dyed [7]. Since then, other studies have mainly focused on comfort and appearance, overlooking important properties such as breaking strength, elongation at break or hydrophobicity [8,9,10,11]. In our previous work [12], we demonstrated the feasibility of using BC, as obtained by static culture, impregnated with two commercial hydrophobic polymers, resulting in a composite with potential textile or leather-like material.

Vegetable oils (VOs) are abundant renewable resources with an increasing number of industrial applications. They offer the advantages of low cost, nontoxicity and biodegradability [13,14,15,16]. Basically, these biopolymers are composed of triglycerides. The fatty acids in most common triglycerides vary from 14 to 22 carbons in length and have 0 to 3 double bonds [15,17,18]. Among the VOs, soybean oil is one of the most attractive due to its low price and abundant availability. To increase their reactivity, double bonds can be replaced by more reactive functional groups such as epoxide, acrylate, hydroxyl or maleate [16,19,20,21]. Most commonly, double bonds are epoxidized and then acrylated, reacting with carboxyl groups of acrylic acids, allowing free radical polymerization [21,22]. Acrylated epoxidized soybean oil (AESO) has been studied extensively in the production of composites with high renewable content. Most of these composites include different particles or fibers, such as microcrystalline cellulose [23], regenerated cellulose fibers [24], coconut waste [25], discarded cotton/polyester and denim fabrics [26,27], ramie fibers [28], hemp fibers [17,29,30], flax and glass fibers [17] or pyrolyzed chicken feather fibers [20].

Regarding leather and analogues, AESO was surface-grafted onto goat leather using UV-radiation [31]. A recent patent [3] demonstrated the possibility of manufacturing ecological leather analogues by mixing natural fibers with epoxidized and acrylated triglycerides and vinyl monomers, the mixture being chemically polymerized. The composite was then deposited in suitable molds and hot pressed. In another work [32], an environmentally friendly leather substitute was developed by reinforcing a mixture of AESO resin with cotton fabrics. Later, an ecological leather composed of organic cotton fabrics and AESO/MLAU (methacrylated lauric acid) (50/50) resin was tested in footwear [33]. Although the authors presented this product as water-resistant and breathable, no tests were performed to support these claims.

Composites of BC with modified soybean oil have been reported. Blaker et al. [34] developed highly porous UV curable nanocomposite foams of BC/AESO by the production of water-in-oil emulsions stabilized by BC nano-fibrils previously hydrophobized by acetylation and silylation. In a more recent work [35], other monomers were added to AESO resin and the emulsions were polymerized by free radicals, for the preparation of thermosetting AESO-BC nanocomposite foams. Lee et al. [36] demonstrated the production of macroporous 3D polymers by microwave heating of gas/AESO liquid foams. The addition of BC allowed a significant improvement of the stability and the mechanical properties. Retegi et al. [37] developed optically transparent composites with excellent mechanical properties when impregnating BC films with epoxidized soybean oil. To improve the dispersion of the nanofibers and the adhesion between the cellulose and the hydrophobic polymer matrix, the BC films were first acetylated. 

From the above, as is the purpose of this work, it is expected that by combining never dried BC membranes, as obtained by microbial fermentation, with a biodegradable polymer such as AESO, it may be possible to produce a truly green nanocomposite with potential applications in the leather industry. BC constitutes a three-dimensional polymeric structure with interconnected fibers, with macroporosity and therefore high aptitude to the anchorage of the AESO emulsified particles. Further, using emulsified VOs as low-cost natural substrates obviates the need to surface-modify BC, thus simplifying the preparation of BC-based composites while not affecting the BC’s native properties. This strategy represents a novel and promising approach towards the development of an environmentally friendly product, exclusively from biological and recyclable materials, at low water and energy production costs. 

## 2. Materials and Methods 

### 2.1. Materials

Bacterial cellulose membranes were offered by Satisfibre S.A. (Braga, Portugal). Soybean oil, epoxidized acrylate (Sigma-Aldrich, Steinheim, Germany), lauryl methacrylate (97%) (Acros Organics, Geel, Belgium), 1,6-hexanodiol diacrylate (80%) (Sigma-Aldrich, Steinheim, Germany), tri(propylene glycol) diacrylate (Sigma-Aldrich, Steinheim, Germany), Triton X-100 (Sigma-Aldrich, Steinheim, Germany), Span 80 (Sigma-Aldrich, Steinheim, Germany), isobutanol (Merck Millipore, Darmstadt, Germany), cumene hydroperoxide (80%) (Sigma-Aldrich, Steinheim, Germany), cobalt naphthenate (6%) (Sigma-Aldrich, Steinheim, Germany), and polyethylene glycol 400 (Merck Millipore, Darmstadt, Germany), were used as received. Persoftal MS Conc.01 and Baygard EFN (Tanatex Chemicals) were offered by ADI Center (Santo Tirso, Portugal). 

### 2.2. Methods

#### 2.2.1. Preparation of the Acrylated Epoxidized Soybean Oil (AESO) Mixture

AESO was used in this work to produce a hydrophobic composite with high bio-based content. AESO is synthesized from soybean oil (renewable resource abundantly available) via epoxidation and acryloilation. It contains three highly reactive functionalities, double (C=C) bonds, –OH groups, and epoxy rings. The C=C bond in AESO is capable of self-polymerizing and copolymerizing with other components via a free-radical initiation, forming a three-dimensional network. However, it has a low crosslinking density and thus inferior mechanical strength due to the existence of long aliphatic chains and low degree of unsaturation in AESO molecules. Also, it has high viscosity at room temperature that restricts its processability [29]. To reduce these limitations, a mixture was prepared by adding different reactive monomers to AESO at room temperature. This mixture was composed of acrylated epoxidized soybean oil (50% m/m); lauryl methacrylate (40% m/m)-a fatty acid-based reactive diluent, potentially bio-based, which reduces the viscosity of the mixture [38]; 1,6-hexanodiol diacrylate (5% m/m); and tri(propylene glycol m/m) diacrylate (5% m/m)-bifunctional monomers which can enhance the crosslinking [39].

#### 2.2.2. Determination of Required the Hydrophilic–Lipophilic Balance (HLB) and AESO Emulsion Stability Evaluation

The Hydrophilic–Lipophilic Balance (HLB) values are generally considered vital for the stabilization of surfactant-based emulsions. Tritons and Spans are a range of non-ionic surfactants stable in mild alkalis, acids and electrolytes and have no reaction with ionic ingredients or actives. To prepare stable AESO emulsions, the effect of HLB was first studied by preparing different combinations of two non-ionic surfactants (Triton X-100 and Span 80) and a co-surfactant (Butanol) in a ratio of 2:1, as presented in Table 1. The HLB values of the mixed surfactants were calculated by Equation (1) [40]: (1)$$HLB_{mix}=\;\left(HLB_{A}X_{A}\right)+\;\left(HLB_{B}X_{B}\right)+\;\left(HLB_{C}X_{C}\right),$$
where *HLB_mix_*, *HLB_A_*, *HLB_B_*, and *HLB_C_* are the HLB values of the mixture, Triton X-100, Span 80, and Butanol, respectively, and *X_A_*, *X_B_* and *X_C_* are the weight percentages of every surfactant in the mixture. HLB values from 5.20 (more lipophilic or oil soluble) up to 11.33 (more hydrophilic or water soluble) were obtained.

The oil-in-water (O/W) emulsions were prepared with a mass ratio of 20:2:78 (AESO mixture/surfactant combination/water), as follows: 2 g of the surfactant combination (Table 1) were added to 20 g of the AESO mixture, followed by the addition of 78 g of deionized water. The mixture was emulsified using a homogenizer (Unidrive X 1000 D, CAT, Staufen, Germany) at a speed of 30,000 rpm, for 1, 5 and 10 min. This process was carried out in an ice bath to avoid temperature rise during emulsification. After this, the emulsions were stored in test tubes at room temperature to investigate their stability over time (up to 10 days) under conditions of varied HLB and stirring time. The stability was evaluated by visually recording signs of phase separation and creaming; the droplet morphology of the emulsions was investigated by optical microscopy using a Leica DM750 M microscope (Leica Microsystems, Wetzlar, Germany) with a Leica MC 170HD camera, using a 10× eyepiece lens and 100× objective lens.

#### 2.2.3. Exhaustion of BC Membranes with Emulsified AESO

To test the incorporation of AESO in BC membranes, an emulsion was prepared by adding the initiator cumene hydroperoxide (CHP) (3% m/m) and the catalyst cobalt naphthenate (CONP) (0.8% m/m) to the AESO mixture. This initiator permits the polymerization of AESO at low temperatures [42], which prevents the aggregation or coalescence of the emulsified AESO particles. An emulsion with HLB of 11.3 (Table 1-D), which provided the best results obtained in the storage stability studies, was prepared using the same procedure described in Section 2.2.2.

BC membranes (with about 3.0 cm in thickness, with a size of 12.0 × 2.5 cm and weighting 90 g) were each treated by exhaustion with 100 g of emulsified AESO mixture for 9 days at 40 °C (Sample 1) followed by a 3 h curing step at 90 °C (Sample 2), to accelerate the cross-link of the emulsified AESO mixture. The composites were then dried at 40 °C in an oven (WTC series, Binder GmbH, Tuttlingen, Germany) for 5 days. To avoid shrinkage of the samples during drying, the composites were attached to a zinc-plated wire support. Regarding the exhaustion process, it was carried out in an Ibelus machine (IL-720, Labelus, Braga, Portugal) equipped with an infrared heating system, using stainless steel cups with a capacity of approximately 220 cm^3^, with a rotation of 50 rpm, 40 cycles, and a temperature gradient of 2 °C·min^−1^. Samples were collected after the exhaustion process and before drying, for analysis by scanning electron cryomicroscopy (described below).

#### 2.2.4. Production of Composites with Different Polymers

Several composites were also produced by adding other polymers to the pre-emulsified AESO mixture. These were PEG 400, Persoftal MS Conc.01 (PDMS-based softener (S)) and Baygard EFN (perfluoroarbon-based hydrophobizer (H)). Softeners (Persoftal) and hydrophobizers (Baygard) used in the textile industry are usually liquid dispersions or emulsions that, in addition to active agents (polysiloxanes or fluorocarbons), contain emulsifiers (e.g., ethoxylated fatty alcohols), dispersants, defoamers. Details on the characteristics of Persoftal and Baygard were presented in our previous work [12]. PEG is a polymer with functionalities such as steric stabilization, which can be used to prevent particle agglomeration [43]; it is a nonionic surfactant able to form long chain structures in aqueous solution; it is also a plasticizer agent with the ability to increase molecular spacing, thus offering flexibility [44].

To produce the composites, BC membranes (with about 3.0 cm in thickness, with a size of 12.0 × 13.0 cm and weighting 450 g) were first squeezed to a final wet mass of 100 g. Then the compressed membranes were each treated by exhaustion with 100 g of an aqueous mixture as shown in Table 2, adding water to complete the 100 g.

For the preparation of these mixtures, the AESO mixture emulsion was first prepared as described on Section 2.2.2 and Section 2.2.3. Then PEG, Persoftal or Baygard were added and the mixture was stirred at 500 rpm for 1 min.

The exhaustion treatment lasted for 5 days at 30 °C, in the same equipment above described, after which the samples were oven dried for 5 days at 40 °C, followed by a curing step for 3 h at 90 °C. As before, to avoid shrinkage of the samples during drying and curing, the composites were attached to a zinc-plated wire support. 

The polymer content in the final composites was calculated through the Equation (2): (2)$$Polymer\;Content=\left(\frac{W_{composite}-\;W_{BC}}{W_{composite}}\right)\times 100$$
where *W_composite_* corresponds to the dry mass of the composite, and *W_BC_* corresponds to the dry mass of BC.

#### 2.2.5. Characterization of the BC-Based Composites

##### Scanning Electron Cryomicroscopy (Cryo-SEM)

SEM analyses of the BC and BC composites were performed using a high-resolution Scanning Electron Microscope (JEOL JSM 6301F, JEOL, Tokyo, Japan) with X-ray Microanalysis (Oxford INCA Energy 350, Oxford Instruments, Abingdon, England) and a CryoSEM (Gatan Alto 2500, Gatan, Pleasanton, CA, USA). The non-dried specimens were rapidly cooled (plunging it into sub-cooled nitrogen-slush nitrogen) and transferred under a vacuum to the cold stage of the preparation chamber. Then the samples were fractured, sublimated (‘etched’) for 120 s at −90 °C, and coated with Au/Pd by sputtering for 45 seconds with a 12 mA current. Afterward, the samples were transferred into the SEM chamber and analyzed at a temperature of −150 °C.

##### Scanning Electron Microscopy (SEM)

Analyses of the surface and cross-section morphology of the dried BC and BC composites were done using an ultra-high-resolution field emission gun SEM instrument (NOVA 200 Nano SEM, FEI Co. Hillsboro, OR, USA). To analyze the cross-section, the samples were first freeze-fractured with liquid nitrogen and coated with a thin layer of Au/Pd. 

##### Fourier Transform InfraRed (FT-IR) Spectroscopy

A Nicolet Avatar 360 FT-IR spectrophotometer (Madison, WI, USA) was used to record the FT-IR spectra of the dried BC sheet and BC composites. The spectra were collected in the attenuated total reflection mode (ATR) at a spectral resolution of 16 cm^−1^, with 60 scans, over the range 650–4000 cm^−1^ at room temperature. A background scan with no sample and no pressure was acquired before the spectra of the samples were collected.

##### Contact Angle (CA) and Surface Free Energy (SFE)

The surface wettability was accessed by contact angle measurements carried out in a DataPhysics instrument (Filderstadt, Germany) with a video system for the capture of images every 0.04 s in static mode using the sessile drop method, using OCA20 software (version 1.5, Dataphysics instrument, Filderstadt, Germany). A drop of 5 µL of distilled water was placed on the dried composite’s surface with a microliter syringe and observed with a special charge-coupled device camera. Afterward, the water contact angle was observed over time for 180 s. At least five measurements at different places were taken for each sample. To calculate the surface energy (*γ_s_*) of the BC composites and their polar (*γ_s_^P^*) and dispersive (*γ_s_^D^*) components, the Wu method (harmonic-mean) was used, with Equation (3) [45]:(3)$$\gamma_{sl}=\;\gamma_{s}+\;\gamma_{l}-4\;\left(\frac{\gamma_{s}^{D}\gamma_{l}^{D}}{\gamma_{s}^{D}+\;\gamma_{l}^{D}}+\;\frac{\gamma_{s}^{P}\gamma_{l}^{P}}{\gamma_{s}^{P}+\;\gamma_{l}^{P}}\right)$$

The following liquids with known surface energy and surface energy components were used: distilled water (*γ*: 72.8; *γ^D^*: 29.1; *γ^P^*: 43.7); polyethylene glycol 200 (*γ*: 43.5; *γ^D^*: 29.9; *γ^P^*: 13.6); and glycerol (*γ*: 63.4; *γ^D^*: 37.4; *γ^P^*: 26.0), units in mJ·m^−2^ [46].

##### Differential Scanning Calorimetry (DSC)

DSC curves were obtained on a Mettler-Toledo DSC822 instrument (Giessen, Germany). Samples weighing about 5–7 mg (the exact mass was recorded before each assay) were heated in hermetically-sealed aluminum pans and tested in the temperature range of 25 to 450 °C at a heating rate of 10 °C·min^−1^, under inert nitrogen atmosphere at 80 mL·min^−1^ flow rate. 

##### ThermoGravimetric Analysis (TGA)

Thermogravimetric analysis was carried out in a Hitachi STA7200 (Tokyo, Japan). Samples weighing 5–7 mg (the exact mass was recorded before each assay) were placed in platinum pans and tested from 25 to 600 °C at a heating rate of 10 °C·min^−1^ under a nitrogen flow rate of 200 mL·min^−1^. 

##### Mechanical Properties

Tensile strength and elongation at break measurements were evaluated according to the standard ISO 17706:2003 [47]. The overall width of the sample (25 mm) was fixed and a length that allows an initial distance between the clamps of the tester equipment (Hounsfield HSK100, Salfords, UK) of 75 mm was set out in grips and subjected to tensile and tear. Three samples of each dried material were tested at a constant speed of 100 mm·min^−1^. 

## 3. Results and Discussion

### 3.1. Emulsions Stability and Diffusion into BC

Both the HLB value of the surfactant mixture (Triton-X100/Span 80/Butanol) and emulsification time had a significant effect on the stability of the O/W emulsions, as evaluated by visual observation over 10 days. As shown in Figure 1b, at HLB of 5.20, the lowest creaming formation was observed in emulsions prepared with longer mixing times. Increasing the HLB value to 11.33, hence increasing the hydrophilic ratio of the surfactant’s mixture, allowed for the stabilization of the emulsion for up to 10 days (Figure 1f), regardless of the emulsification time, as no creaming was observed after 10 days of storage. 

The optical microscopy images of the emulsions prepared with different HLB values were taken immediately following preparation. At lower HLB (Figure 1a,c), emulsions exhibit larger droplet size (possibly containing also some multivesicular droplets, Figure 1a) and size variability, as opposed to those with higher HLB. Thus, a surfactants mixture with HLB 11.33 and 10 min of emulsification was selected for further work.

After determining the best HLB value for the preparation of the AESO emulsions, impregnation of BC was done through the exhaustion process. This process commonly used in textile technology involves placing the fabric or yarn in a chamber containing water and treatment products. The chamber is then sealed and the treatment solution heated and submitted to heavy stirring, which results in the products transitioning from the water to the fabric or yarn. This procedure was adopted in this work to process the BC membranes, incorporating the AESO emulsion into the cellulosic porous network. Although several authors addressed the development of composites using BC fibers, very few papers use the intact membranes obtained by static fermentation, as it is demonstrated in this work. To confirm the presence of AESO particles inside the BC 3D nanofibrillar matrix, the composite samples were examined by Cryo-SEM. Figure 2 shows the 3D porous fiber network structure of BC (Figure 2a) and of the AESO particles adsorbed onto the surface of the nanofibers (Figure 2b), after 9 days at 40 °C, showing no signs of particle aggregation or coalescence. However, by increasing the temperature to 90 °C for 3 h, during the exhaustion process (Figure 2c), the coalescence of the AESO particles was observed, possibly due to the emulsion breakdown at a higher temperature and to lower viscosity, associated with an incomplete polymerization. Figure 2d shows a SEM micrograph of the dried BC/AESO composite at a higher magnification. The collapse of the BC porous structure is observed upon drying, producing a structure with packed AESO particles. Further work was performed by drying the material at 40 °C before post curing at 90 °C, attempting to control the coalescence to some extent.

### 3.2. Morphological Analysis

BC-based composites were also obtained through exhaustion with AESO emulsion combined with other polymers. The morphological properties of the dried BC and BC-based composites were evaluated by SEM (Figure 3). As expected, the surface and cross-section of dried BC (Figure 3a,e) presented a compacted structure of nanofibers, due to the replacement of cellulose–water–cellulose by cellulose–cellulose hydrogen bonding, the nanofibers being randomly arranged in layers with pores on the surface and throughout the BC matrix [48]. The cross-section SEM analysis of the dried BC composites (Figure 3f–l) confirmed a good bulk penetration of the emulsified polymers during the exhaustion process, coating the BC nanofibers and promoting a more compact and bulky structure, as compared to native BC. SEM images of the surface of the BC composites (Figure 3b–d) show that the polymers also covered the membrane’s surface, resulting in a rougher structure where the nanofibers are not distinguished. While sphere-like particles can be observed in cross-section images, the possibility of particle coalescence during drying cannot be excluded. Despite the complete coverage of the fibers by the polymers, the characteristic stratified BC structure is still observed in some cases (ex. BC/AESO: Figure 3f, and BC/AESO/PEG/H: Figure 3h).

As compared to BC, the composites presented an increase in thickness and in the mass per unit area, due to the incorporation of the polymers (Table 3), which varied with the formulation. As compared to BC impregnated with AESO mixture, both the added softener (S, Persoftal) and hydrophobizer (H, Baygard) presented lower impregnation. However, the opposite effect is observed adding PEG to either S, H or both mixtures, suggesting that PEG promoted a higher BC impregnation content. In the case of the mixtures of AESO, S and/or H, it is not possible to determine whether these were incorporated into BC in the same proportions as that of the emulsified AESO.

### 3.3. FTIR Analysis

Fourier transform infrared (FTIR) spectroscopy was used to characterize the functional groups on the dried BC and BC composites’ surfaces (Figure 4). BC spectrum exhibited the characteristic cellulose vibration peaks, namely, –OH stretching peak at 3344 cm^−1^, C–H stretching at 2919 cm^−1^, –OH bending at 1650 cm^−1^, –CH_2_– bending at 1426 cm^−1^, C–O–C deformation modes and stretching vibrations at 1159–1107 cm^−1^, C–O–C and C–OH stretching vibration of the sugar ring at 1054–1029 cm^−1^, and C–OH out-of-plane bending mode at 665 cm^−1^ [49,50,51].

The spectrum of the BC/AESO composite showed a peak at 3343 cm^−1^ corresponding to the stretching vibrations of –OH groups. The peaks at 2922–2855 cm^−1^ and at 1457–1406 cm^−1^ are attributed to the asymmetric stretching vibrations and deformation of C–H in the –CH_2_– and –CH_3_ bonds, respectively, assigned to the inherent aliphatic sequences of AESO. Another significant peak at 1721 cm^−1^ is attributed to the stretching vibration of C=O in esters and the one at 1637 cm^−1^ is ascribed to the double-bond signals of acrylate functionalities (–CH=CH_2_). The peaks at 1295 cm^−1^ and 1163–1053 cm^−1^ correspond to C–O groups and C–O–C stretching vibration of ester, respectively. Finally, the peak at 811 cm^−1^ corresponds to the bending vibration of =C–H, the double bonds on AESO characteristic of the epoxide group [23,52,53].

In the BC/AESO/PEG composite, the increase of some absorption bands can be attributed to PEG400, although they overlap with signals from cellulose and AESO mixture. These bands are located at 3345 cm^−1^ (–OH stretching), 2918–2854 cm^−1^ (CH_2_ stretching), 1644 cm^−1^ (–OH bending), 1456 cm^−1^ (asymmetric CH_2_ deformation), 1351 cm^−1^ (CH_2_ wagging), 1095–1029 cm^−1^ (CH_2_ symmetric deformation, C–O–C and C–OH stretching), 950 cm^−1^ (CH_2_ rocking), and 664 cm^−1^ (C–OH out-of-plane bending) [43,54]. 

In the composite BC/AESO/S, the appearance of new peaks was observed, namely, at 1258 cm^−1^ (CH vibration in Si–CH_3_) and at 791 cm^−1^ (NH_2_ and Si–CH_3_), confirming the incorporation of modified amino-PDMS into the BC composite [55,56,57]. When PEG 400 was used in the polymer mixture, it was also observed at the peak at 1575 cm^−1^ that is attributed to the vibration modes of NH_2_ groups [58]. The vibration bands of Si–O–C and Si–O–Si bridges at around 1165–1011 cm^−1^ are difficult to analyze as they overlap with the C–O–C vibrations from BC and PEG [59,60].

In the composites with perfluorcarbon (for BC/AESO/PEG/H), it was possible to identify the bands associated with CF_2_ groups (asymmetric and symmetric CF_2_ stretching at 1234 and 1141 cm^−1^ respectively and ‘amorphous’ CF_2_ deformations at 702 cm^−1^) [61]. While in the composite BC/AESO/PEG/S/H, perfluorocarbon peaks are difficult to identify as they overlap with the characteristic peaks of the PDMS-based polymer.

### 3.4. Surface Wettability and Surface Free Energy

The BC composites are designed for leather (textile and footwear) applications; therefore, it is important to determine their surface hydrophobicity. The wetting properties of the BC and BC-based composites were investigated by measuring the water contact angles (WCAs). The obtained values are shown in Figure 5a,b. BC has a highly hydrophilic surface, bearing the lowest water droplet angle (63.1°), which increased for the BC composites to values between 79.0° and 138.0°, indicating a significant increase in hydrophobicity. Values of 95.8° and 79.0° were observed for BC/AESO and BC/AESO/PEG, respectively. AESO contains hydrophobic long-chain non-polar fatty acid chains [62] and consequently improves the water resistance of the composite. Despite being more hydrophobic, the WCAs over time decreased quickly in these composites, as compared to BC. This can be explained by the closed packed structure of the dried BC that limits the water diffusion through the tight space between the nanofibers, due to the strong and high number of cellulose–cellulose hydrogen bonds [50]. As expected, the incorporation of PDMS (S) and perfluorocarbon-based (H) polymers into BC also significantly increased the WCAs. Both polymers have very low surface tension, 19.5–23.6 mN·m^−1^ [63,64] and 6–18 mN·m^−1^ [65], respectively, thus decreasing the free energy of the system and reducing the surface wettability. However, in the samples with PEG, this increase was not as substantial, since PEG can interact with the BC membrane by hydrogen bonds. As stated by Kondo et al. [66], the ether oxygen in the poly(ethylene oxide) skeleton forms hydrogen bonds with the primary OH group at the C6 position of the anhydroglucose unit. Promising results were observed with all H-composites: higher contact angles (Figure 5b) and very low absorption rate over time (Figure 5a). The increase in hydrophobicity resulted from a decrease in the polar component of the surface tension and consequently in the total surface free energy of the composite, as shown in Figure 5b. An exception to this trend was observed for BC/AESO/PEG, where an increase in the surface free energy was recorded. Among the different polymers tested, AESO was the one that offered less hydrophobicity, and the incorporation of the PEG, being hydrophilic as observed above, increased the value of the polar component. 

### 3.5. Thermal Properties

The thermal properties were evaluated by differential scanning calorimetry (DSC) and thermogravimetric analysis (TGA). DSC and TGA curves of BC and its composites obtained under nitrogen atmosphere are depicted in Figure 6a,b, respectively, and the corresponding relevant data are summarized in Table 4.

The DSC curve of BC reveals an endothermal degradation peak (T_m_) at 362.2 °C, which is attributed to partial pyrolysis with the fragmentation of carbonyl and carboxylic bonds from anhydrous glucose units [67]. BC also shows a narrow weight loss at 351.0 °C (T_dmax_), indicating fast degradation, involving dehydration, depolymerization of the main polymer network and the decomposition of glucosyl units followed by the formation of a charred residue [68,69].

In the case of the composites, an additional endothermic transition below 100 °C is observed in the DSC curves, it corresponds to the first step of weight loss in the temperature range of 30–150 °C observed in Figure 6b, which is ascribed to the loss of absorbed water. From Table 4, it is possible to observe that the composites with PEG 400 lost 5% of their mass at lower temperatures; this occurs because low-molecular polyethylene glycol contributes to higher hygroscopicity, the water being released by evaporation. These samples also present less charred residue, when compared with the composites without PEG. For most of the composites, a second event, in the range of 100–250 °C, is observed before the main degradation. This degradation stage can be considered the evaporation and decomposition of unreacted monomers, catalysts, or other low molecular weight components in the composites [22,70,71,72].

All DSC curves of the composites display exothermic transitions up to 200 °C, which we hypothesize to correspond to the polymerization of unreacted AESO [73]. A redox initiator system (the initiator cumene hydroperoxide (CHP) and the promoter cobalt naphthenate (CONP)) was used to polymerize AESO during the exhaustion process. This was performed at a relatively low temperature (30 °C), as described by other authors [42], attempting to avoid coalescence of the emulsified AESO (which is favored at a higher temperature due to the reduction of viscosity). However, as stated by Dweib et al. [42], oxygen and water inhibit the free-radical polymerization reaction and the complete curing of the resins. In this work, emulsified AESO was cured, probably not completely, in aqueous media. Thus, the exothermic peak in DSC curves may be associated with the free-radical polymerization. Indeed, this peak does not appear in the AESO mixture curve tested without the addition of a catalyst on Figure 6a. The increase in temperature promoted the decomposition of the initiator CHP, generating more free-radicals to complete the polymerization. Solutions to achieve full polymerization will be reported in forthcoming work.

Further, the degradation of the composite network structure constitutes the second endothermic transition at higher temperatures. The beginning of the structural disruption is defined as the temperature where 10 wt% of the mass is lost. It was observed that T_10wt%_ for BC (326.4 °C) was higher than T_10wt%_ for all composites. Therefore, it can be inferred that native BC is more thermally stable than the polymeric composites. 

The composites show less pronounced peaks with the main mass loss step, corresponding to the higher percentage of mass loss, observed in a broader temperature range. The T_dmax_ decreases in the order BC > BC/AESO > BC/AESO/PEG > BC/AESO/S > BC/AESO/PEG/S > BC/AESO/PEG/S/H > BC/AESO/H > BC/AESO/PEG/H and these results are in accordance with the DSC data. Although the composites are less stable thermally, they are stable up to 200 °C, so they can be applied in common leather applications.

### 3.6. Mechanical Properties 

The average values and standard deviation of the tensile strength and elongation at break of the BC and BC-based composites are reported in Table 5, and the stress–strain curves are given in Figure 7. As deduced from Figure 7, neat BC displayed a rigid and brittle behavior, because of the extensive interactions between the polymer molecules that result in high tensile strength but low elongation at break. The incorporation of the plasticizers into polymers disrupts the intermolecular attractive forces between the main polymer chains and consequently increases the free volume and chain mobility, leading to an increase in extensibility [50,74,75]. Thus, when compared to the neat BC membrane, the incorporation of emulsified polymer mixtures leads to a reduction in the values of the tensile strength and an increase in the elongation at break. As observed by SEM (Figure 3), the polymers appeared to have completely covered the surface of the BC nanofibers, resulting in an increase in the composites’ thickness (Table 3). As observed by DSC, AESO was not completely polymerized, thus it may have not fully acted as a reinforcing agent but also as a plasticizer, allowing higher mobility between the different layers of the BC membrane. 

The addition of either S or H, to BC/AESO, improved the mechanical and elongation properties of the composites, as compared to BC/AESO. However, only for this composite (BC/AESO) the addition of PEG allowed increasing the mechanical properties. In general, it was observed that the addition of different polymers leads to a decrease in tensile strength and an increase in elongation at break. The incorporation of PEG resulted in a consistent and significant increase in the elongation at break values due to its plasticizing effect.

Overall, taking as reference the technical report ISO/TR20879 [76], with respect to the mechanical properties, most of the composites are suitable to be used in footwear. Regarding tensile strength values, BC/AESO/PEG/S, BC/AESO/PEG/H, and BC/AESO/PEG/S/H samples (more polymer content as shown in Table 3) were below the reference value (casual footwear: >10 N/mm) (MPa × Thickness), but in the case of the elongation at break, all samples presented values within the required reference values.

## 4. Conclusions

This work provided a straightforward method to prepare BC composites with high potential for applications as a replacement for leather. We have successfully prepared composites based on BC, emulsified AESO resin, PEG, and PDMS- and perfluorocarbon-based polymers through a simple strategy to enhance the flexibility and hydrophobicity of the BC.

Based on SEM observations and FTIR analysis, all the tested polymers penetrated well and uniformly into the BC matrix. The obtained composites showed hydrophobicity with the highest values of WCAs obtained for the composites with the perfluorocarbon-based product. Regarding the thermal and mechanical properties, it was found that the composites presented lower thermal stability and tensile strength, although they are stable up to 200 °C and most of the composites can be applied in uppers for shoes.

Further optimization of the process may improve its performance through improved control of the polymerization reaction. Hence, this work opens new perspectives for potential applications of BC in the footwear industry.

## Figures and Tables

**Figure 1 nanomaterials-09-01710-f001:**
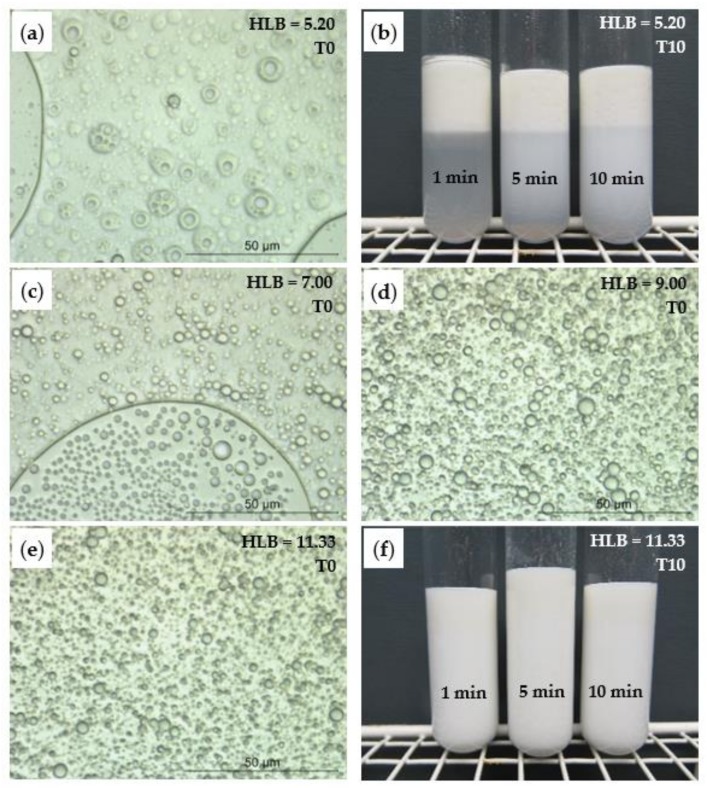
Optical micrographs (100× magnification) of freshly prepared AESO emulsions with different HLB and after 10 min of emulsification: (**a**) HLB 5.20, (**c**) HLB 7.00, (**d**) HLB 9.00, and (**e**) HLB 11.33; and visual appearance of AESO emulsions at different times of emulsification after 10 days storage: (**b**) HLB 5.20 and (**f**) HLB 11.33.

**Figure 2 nanomaterials-09-01710-f002:**
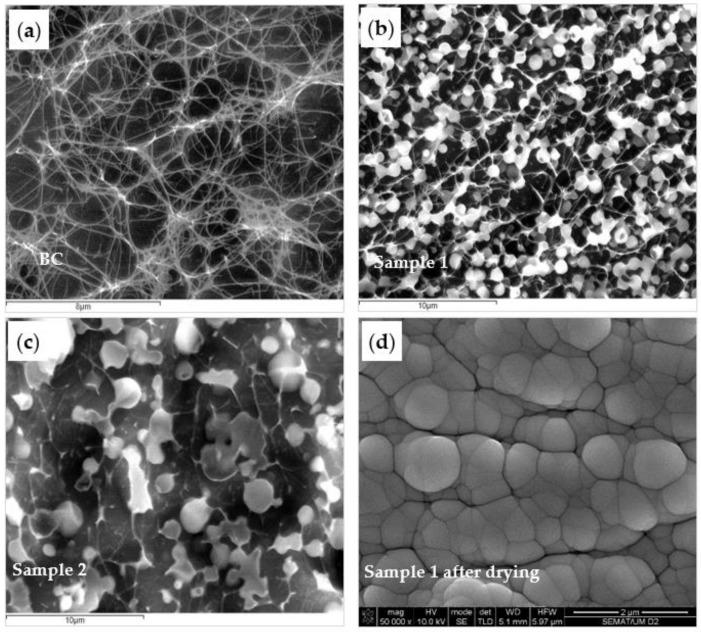
CryoSEM images: (**a**) BC membrane, (**b**) after exhaustion with AESO emulsion for 9 days at 40 °C (Sample 1), (**c**) after exhaustion with AESO emulsion for 9 days at 40 °C + 3 h at 90 °C (Sample 2); and (c) SEM image of the Sample 1 after drying at 40 °C.

**Figure 3 nanomaterials-09-01710-f003:**
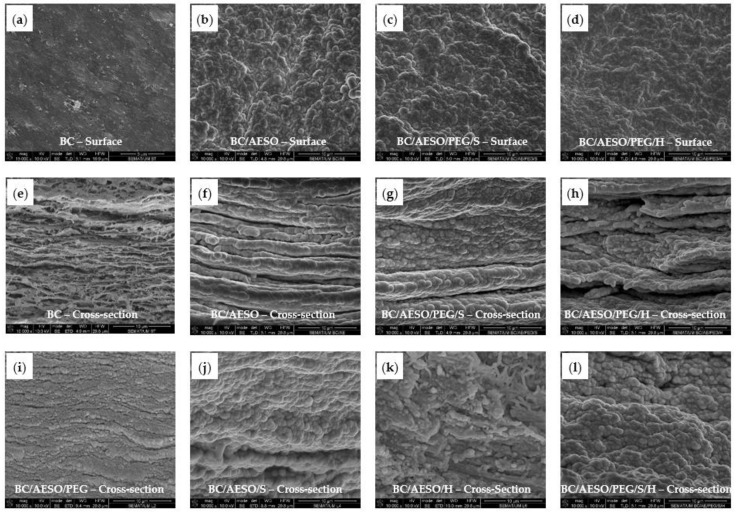
SEM images of BC and BC composites: surface and cross-section images, (**a**,**e**) BC; (**b**,**f**) BC/AESO; (**c**,**g**) BC/AESO/PEG/S; (**d**,**h**) BC/AESO/PEG/H; and cross-section images (**i**) BC/AESO/PEG, (**j**) BC/AESO/S, (**k**) BC/AESO/H, and (**l**) BC/AESO/PEG/S/H. Magnification: 15,000× (scale: 5 µm) (a) and 10,000× (scale: 10 µm) (b–**l**).

**Figure 4 nanomaterials-09-01710-f004:**
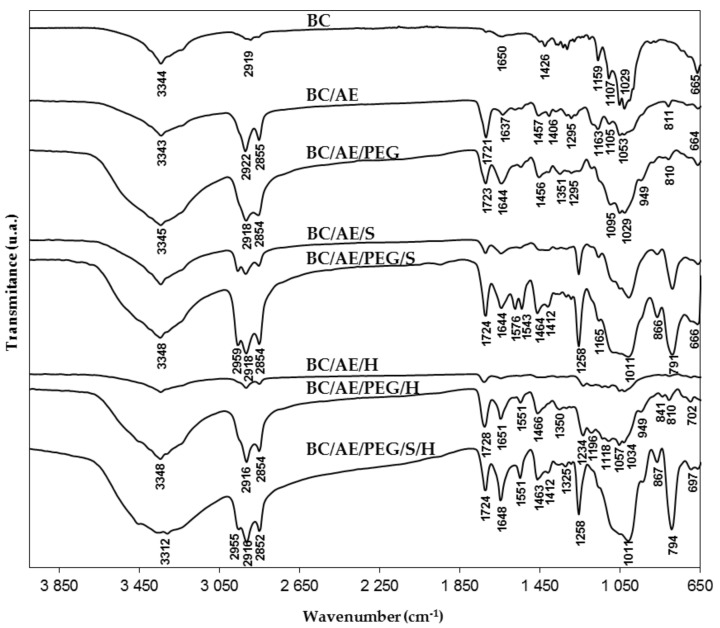
FTIR spectra of BC and the composites.

**Figure 5 nanomaterials-09-01710-f005:**
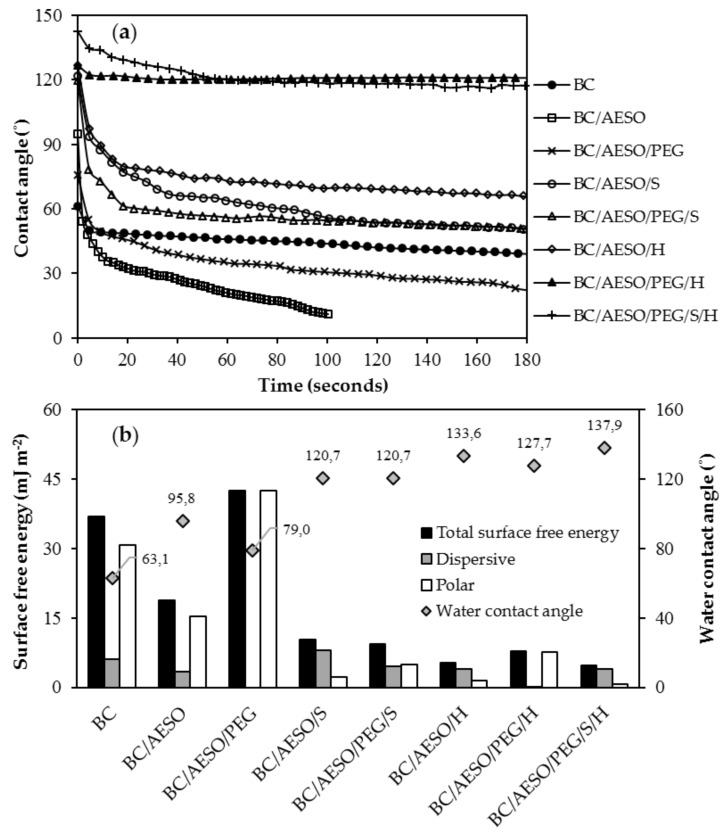
(**a**) Water contact angle over time and (**b**) surface free energy.

**Figure 6 nanomaterials-09-01710-f006:**
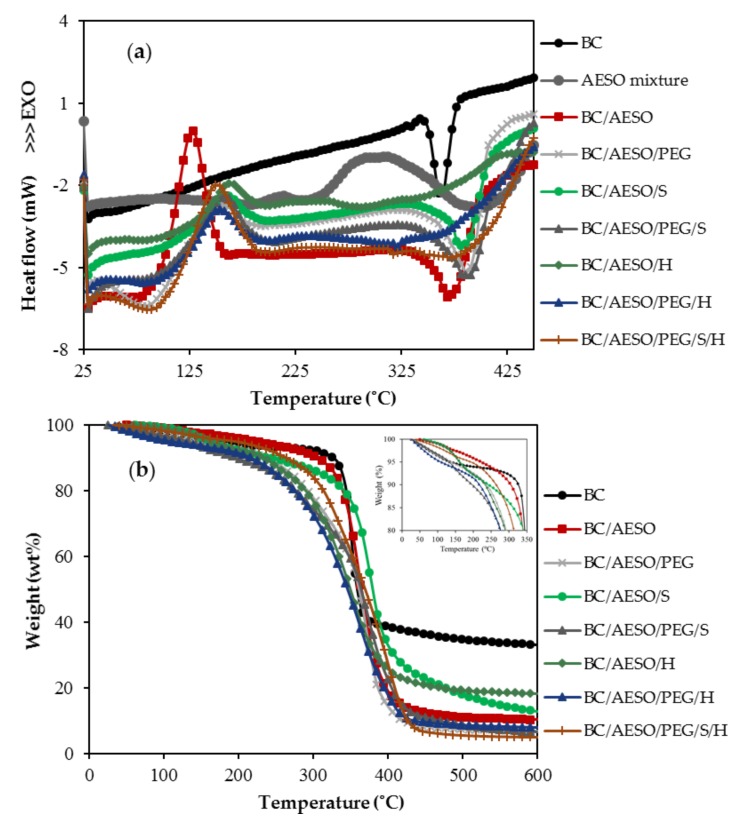
(**a**) DSC thermograms and (**b**) TGA curves of weight percentage of dried BC and BC composites.

**Figure 7 nanomaterials-09-01710-f007:**
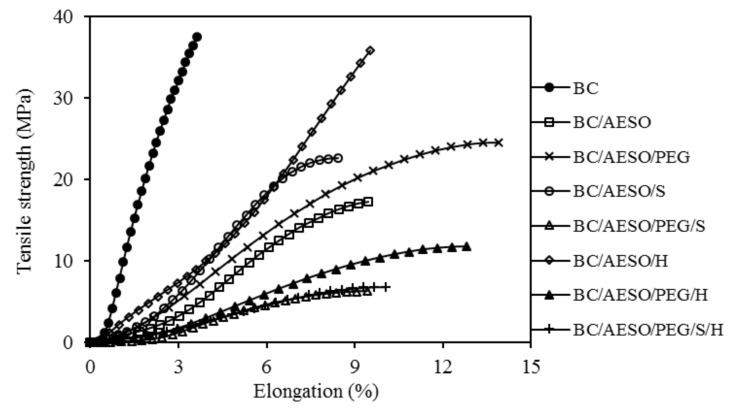
Stress strain curves of dried BC and BC composites.

**Table 1 nanomaterials-09-01710-t001:** Combinations of surfactants used to study the effect of HLB on the AESO emulsion stability.

Surfactant Combination	Triton X-100 (%) (HLB = 13.5) ^1^	Span 80 (%) (HLB = 4.3) ^1^	Butanol (%) (HLB = 7.0) [41]	HLB_mix_
A	0.00	66.67	33.33	5.20
B	19.57	47.10	33.33	7.00
C	41.31	25.36	33.33	9.00
D	66.67	0.00	33.33	11.33

^1^ Values taken from the technical specification sheets

**Table 2 nanomaterials-09-01710-t002:** Proportions of the polymers in the aqueous mixture used in the production of BC composites.

Sample	AESO Emulsion (g)	PEG 400 (g)	Persoftal MS (g)	Baygard EFN (g)
BC/AESO	75	-	-	-
BC/AESO/PEG	75	4.5	-	-
BC/AESO/S	75	-	18	-
BC/AESO/PEG/S	75	4.5	18	-
BC/AESO/H	75	-	-	18
BC/AESO/PEG/H	75	4.5	-	18
BC/AESO/PEG/S/H	75	4.5	9	9

**Table 3 nanomaterials-09-01710-t003:** Thickness, mass per unit area and polymer content of the composites.

Sample	Thickness (mm)	Mass Per Unit Area (g·m^−2^)	Polymers Content (%)
BC	0.48	238.0	-
BC/AESO	0.78	686.4	65.3
BC/AESO/PEG	0.55	576.0	58.7
BC/AESO/S	0.51	552.0	56.9
BC/AESO/PEG/S	0.79	921.6	74.2
BC/AESO/H	0.48	478.4	50.3
BC/AESO/PEG/H	0.72	886.4	73.1
BC/AESO/PEG/S/H	0.95	1032.5	76.9

**Table 4 nanomaterials-09-01710-t004:** Thermal degradation data obtained from DSC, TGA and DTG curves of dried BC and BC composites.

Sample	T_c_ (°C) ^a^	T_m_ (°C) ^b^	T_5 wt%_ ^a^ (°C) ^c^	T_10 wt%_ ^b^ (°C) ^d^	T_dmax_ ^d^ (°C) ^e^
BC	-	362.2	136.1	326.4	351.0
BC/AESO	127.3	365.0	221.9	305.0	364.4
BC/AESO/PEG	153.2	387.2	136.8	236.8	372.7
BC/AESO/S	158.1	387.6	161.9	242.9	375.9
BC/AESO/PEG/S	157.3	389.9	132.5	196.9	374.3
BC/AESO/H	161.8	350.2	161.4	232.9	353.1
BC/AESO/PEG/H	151.9	365.5	109.5	217.4	361.9
BC/AESO/PEG/S/H	150.1	388.4	192.4	263.8	333.8/400.8

^a^ Temperature of the curing exotherm maximum; ^b^ Temperature of the degradation endothermal maximum; ^c^ Temperature at 5% mass loss; ^d^ Temperature at 10% mass loss; and, ^e^ Temperature(s) at the maximum mass loss rate.

**Table 5 nanomaterials-09-01710-t005:** Tensile strength and elongation at break of dried BC and BC composites.

Sample	Tensile Strength (MPa)	Elongation at Break (%)
BC	37.5 ± 0.8	3.6 ± 0.6
BC/AESO	17.3 ± 0.7	9.5 ± 1.0
BC/AESO/PEG	24.6 ± 1.7	13.9 ± 0.8
BC/AESO/S	22.6 ± 3.3	8.4 ± 0.1
BC/AESO/PEG/S	6.3 ± 1.0	9.4 ± 2.3
BC/AESO/H	35.9 ± 3.0	9.5 ± 0.2
BC/AESO/PEG/H	11.8 ± 1.0	12.9 ± 2.0
BC/AESO/PEG/S/H	6.9 ± 0.2	10.0 ± 0.5

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
