# Peer review of "Bacterial Cellulose and Emulsified AESO Biocomposites as an Ecological Alternative to Leather"

_nanomaterials, 2019, doi:10.3390/nano9121710_

Round 1
Reviewer 1 Report
accept as it is.
Reviewer 2 Report
In this work, the composites based on bacterial cellulose (BC), emulsified acrylated epoxidized soybean oil (AESO), polyethylene glycol, and polydimethylsiloxane- and perfluorocarbon-based polymers were prepared and investigated. The authors declare the high potential of the developed composites for applications as a replacement for leather.
My comments and questions for the authors:
The selection of the AESO mixture composition (Chapter 2.2.1) should be described in more details. Why each comonomer (lauryl methacrylate, hexanodiol diacrylate, and tri(propylene glycol) diacrylate) was selected for the AESO mixture? How the required amount of each comonomer in the AESO mixture was determined or selected? How the proportions of the polymers in the aqueous mixtures used in the production of BC composites (Table 2) were selected? Explanation is needed. How the end of each composite preparation step (exhaustion with emulsified AESO mixture, thermal curing, and drying in oven) was determined? Explanation is needed. Was the mechanical testing of the composite specimens performed in accordance to ISO/TR20879? If yes, it should be written in the methodical part. If no, the comparative tests with the commercial textile and footwear material specimens should be performed at the same experimental conditions. The results of the investigation of BC composites showed that there can be the rest of monomers, catalyst, etc. after their preparation. The low molar mass compounds could migrate out of the composites during the contact with different liquids (water, sweat, etc.). As the potential application of the developed composites is in textile and footwear, it is important to determine the amount of the extractable materials. The biodegradation of natural oil-based polymers depends on their structure (used comonomers, cross-linking density, etc). How long would maintain the mechanical and other properties of the developed composites?
Reviewer 3 Report
This research provides the possibility of replacing microbially produced bacterial cellulose (BC) with leather products, which is very meaningful from the viewpoint of animal protection and the utilization of materials utilizing microbial substance production. From the viewpoint of using BC, which is a nanofiber network, it is considered to be consistent with the purpose of this journal "Nanomaterials". The current manuscript shows many fundamental data of BC-AESO composites. However, I think that the following modifications are necessary to publish.
L287: The authors described that Despite the complete coverage of the fibers by the polymers, the characteristic stratified BC structure is still observed in some cases (Fig. 3d,h). The thickness of the layered structure in the BC composites are thicker than that of the original BC. Does that mean that the gap between the layers has increased? Please explain the reason.
Fig.4: transmittance peaks of the FTIR spectrum from neat BC are too weak. Is it possible to show the peaks more clearer?
L356: the authors describe hydrogen bonds between PEG and BC membrane. Does "numerous hydroxyl groups" mean those derived from the terminal moieties of PEG 400 chains? Is there any other interaction working? For example, Kondo et al. proposed that ether oxygen in the PEO chains engage in a strong hydrogen-bonding interaction with C-6 positioned OH groups of the cellulose molecule.
Round 2
Reviewer 2 Report
The manuscript was improved according to my comments.
Author Response
Thank you for your positive comments.